biomedical engineering/spectroscopy

alginate, calcium carbonate, extracellular matrix mineralization, hydrogel scaffolds, light spectroscopy, tissue engineering

**Author for correspondence:**
J. Lovecchio
e-mail: joseph.lovecchio@unibo.it

[†]These authors contributed equally.
[‡]Present address: Department of Industrial Engineering (DIN), University of Bologna, Bologna (BO), Italy.

# Design of a custom-made device for real-time optical measurement of differential mineral concentrations in three-dimensional scaffolds for bone tissue engineering

J. Lovecchio[1,†], V. Betti[1,†,‡], M. Cortesi[2], E. Ravagli[3], S. Severi[1,2] and E. Giordano[1,2,4]

[1]Laboratory of Cellular and Molecular Engineering 'Silvio Cavalcanti'—Department of Electrical, Electronic and Information Engineering 'Guglielmo Marconi' (DEI), University of Bologna, Cesena (FC), Italy
[2]BioEngLab, Health Science and Technology, Interdepartmental Center for Industrial Research (HST-CIRI), Alma Mater Studiorum—University of Bologna, Ozzano Emilia (BO), Italy
[3]Department of Medical Physics and Biomedical Engineering, University College London, UK
[4]Advanced Research Center on Electronic Systems (ARCES), University of Bologna, Bologna (BO), Italy

JL, 0000-0002-4721-1388; MC, 0000-0002-3731-7760; EG, 0000-0002-8054-2875

Monitoring bone tissue engineered (TEed) constructs during their maturation is important to ensure the quality of applied protocols. Several destructive, mainly histochemical, methods are conventionally used to this aim, requiring the sacrifice of the investigated samples. This implies (i) to plan several scaffold replicates, (ii) expensive and time consuming procedures and (iii) to infer the maturity level of a given tissue construct from a cognate replica. To solve these issues, non-destructive techniques such as light spectroscopy-based methods have been reported to be useful. Here, a miniaturized and inexpensive custom-made spectrometer device is proposed to enable the non-destructive analysis of hydrogel scaffolds. Testing involved samples with a differential amount of calcium salt. When compared to a reference standard device, this custom-made spectrometer demonstrates the ability to perform measurements without requiring elaborate sample preparation and/or a complex instrumentation. This preliminary study shows the feasibility of light spectroscopy-based methods as useful for the

non-destructive analysis of TEed constructs. Based on these results, this custom-made spectrometer device appears as a useful option to perform real-time/in-line analysis. Finally, this device can be considered as a component that can be easily integrated on board of recently prototyped bioreactor systems, for the monitoring of TEed constructs during their conditioning.

## 1. Introduction

Scaffold-based tissue engineering (TE) aims at providing cells with a three-dimensional architecture that mimics the native extracellular matrix (ECM). In particular, bone ECM is involved in granting adequate mechanical support, and in regulating cell adhesion, colonization, migration, proliferation and differentiation, to provide the functional characteristics of the mature tissue [1]. To address bone tissue engineered (TEed) constructs towards a suitable maturity level, they are often conditioned using appropriate bioreactor systems [2–5] to administer mechanical stimuli that are known to play a key role in bone tissue mineralization. Monitoring maturation of bone TEed constructs in these settings is important to ensure the quality of applied protocols. In this respect, *in-silico* models, aiming at improving the characterization of three-dimensional cell cultures [6,7] have also been proposed. However, mainly histochemical, destructive methods are currently used, requiring the sacrifice of the investigated samples. This implies the need for planning several scaffold replicates, leading to expensive and time consuming procedures that also infer the maturity level of a given tissue construct from a cognate replica. Non-destructive in-line analysis would help to solve these limits. Indeed such an approach is the state of the art of the clinical assessment of bone tissue remodelling with a dual-energy X-ray absorptiometry (DXA)-based measurement. This test conveniently reports on bone mineral content (BMC) and density (BMD) in patients, although requiring dedicated expensive and bulky instruments [8] inappropriate for a cellular study in a laboratory. Innovative methods are thus catching on. Among those, X-ray μCT was demonstrated as a powerful non-destructive imaging technique to obtain, with the highest resolution, precise quantitative and qualitative information on the bone tissue microstructure, again without the need for conventional destructive methods [9]. However this technique might be limiting due to its still-elevated costs. Light spectroscopy-based methods have been reported to be useful to investigate bone ECM features and might offer an approach for a non-destructive monitoring of TEed constructs maturation. Among these, elemental compositional analysis has been performed using Fourier-transform infrared spectroscopy (FTIR), to confirm apatitic carbonate and phosphate within sample matrices [10]. Similarly, Raman spectroscopy has been used to detect mineral components during osteogenesis [11], including amorphous calcium phosphate (ACP) [12], octacalcium phosphate (OCP) [13], hydroxyapatite (HAP) [14,15], β-tricalcium phosphate (β-TCP) and dicalcium phosphate dehydrate (DCPD) [16]. However, these approaches again require elaborate sample preparation and/or expensive/complex instrumentation.

In this work, we thus explored the reliability of measuring differential amounts of calcium salt in an alginate hydrogel—that is reported as a promising biopolymer for bone TE applications [17], by using a miniaturized and inexpensive custom-made spectrometer device. This original device would enable real-time, in-line, *in situ* analysis, such as is desirable in a bioreactor system. To the best of our knowledge, this is the first account of an inexpensive and miniaturized system capable of non-destructive characterization of the mineral content in polymeric scaffolds. As such, it holds great potential for the monitoring of bone TE scaffolds during their maturation process.

## 2. Material and methods

### 2.1. Three-dimensional alginate hydrogel scaffolds

Three-dimensional alginate hydrogel scaffolds were obtained using the internal gelation polymerization method as described in [18]. Briefly, sodium alginate powder (Sigma-Aldrich, St Louis, MO, USA) was sprinkled on an aluminum foil and sterilized within the laboratory hood through its exposure to UV light for 1 h. The sterilized powder was mixed with sterile RPMI 1640 medium to produce a 2% alginate solution.

Eppendorf tubes (2 ml) were used as scaffold moulds in upside down position. Their cap was wrapped with a double parafilm layer to form a flat surface, while their tip was pierced to allow the casting of the solution.

**Figure 1.** Three-dimensional alginate hydrogels. Three scaffolds are shown at different CaCO$_3$ concentrations of 20, 40 and 80 mM (from left to right).

**Table 1.** Three-dimensional alginate hydrogel scaffold reagent mix.

|  | 20 mM | 40 mM | 80 mM |
|---|---|---|---|
| 2% alginate | 270 µl | 270 µl | 270 µl |
| CaCO$_3$ | 80 µl (135 mM) | 80 µl (270 mM) | 80 µl (540 mM) |
| RPMI 1640 | 128.75 µl | 236.25 µl | 65 µl |
| GDL (1 M) | 21.25 µl | 42.5 µl | 85 µl |

Each mould was filled with 500 µl of final volume at the desired CaCO$_3$ concentration, obtained as reported in table 1.

2% alginate, CaCO$_3$ and RPMI 1640 medium were mixed within each mould vortexing for 10 s. Gluco-delta-lactone (GDL, Sigma-Aldrich) was then added and 5 s 'touch and go' vortexing was applied twice to trigger polymerization.

Scaffolds were left to polymerize overnight at 37°C and 5% CO$_2$. The day after, they were extracted from their moulds, washed with 0.1 M Hepes buffer and maintained in 2 mL RPMI medium in 24 well cell culture plates. Medium was replaced every 48 h. Final scaffold dimensions were 6 × 5 mm, H×.

In each experiment, a set of three-dimensional alginate hydrogel scaffolds, produced with different CaCO$_3$ levels (20, 40, 80 mM; (figure 1; electronic supplementary material, figure S2), was measured at different time points (0, 4, 7, 11 days), to validate the custom-made spectrometer device. $N = 3$ samples for each CaCO$_3$ level and time point (a total of 36 samples) were analysed. Three independent experiments were performed, for a total of 108 measurements.

## 2.2. Custom-made spectrometer device development and measurements

A custom-made device (figure 2) was prototyped and used as a compact and portable spectrometer (40 × 45 × 45 mm, l × W × H) to perform optical measurements. It consists of: a LED array emitter (figure 2a) incorporating five LED chips (MTMD6788594SMT6, Marktech Optoelectronics, NY, USA), a sensing element (figure 2b) based on a mini-spectrometer integrating MEMS (measurement range 640–1000 nm with a step of 20 nm) and image sensor technologies (C11708MA, Hamamatsu Photonics KK, Hamamatsu City, Shizuoka), and a dedicated data acquisition board (figure 2c) (C14465, Hamamatsu Photonics KK). A custom-made scaffold holder (figure 2d) was three-dimensionally printed with a transparent polymer (Biomed Clear Resin, Formlabs, Somerville, MA, USA) and used to maintain the scaffold in a fixed position and distance from the mini-spectrometer, also allowing the easy replacement of the sample at each measurement. A dedicated software (HMSEvaluation, Hamamatsu Photonics KK) was used to acquire the data.

The HMSEvaluation software collects the transmittance data acquired from the scaffold. To this aim, a list of calibration coefficients and parameters had to be set. In this work, the calibration coefficient default values, an exposure time of 5 ms, and the average of 10 acquired spectra for each sample were used.

To avoid interference from the external ambient light, each measurement was performed within a dark casing. Once acquired, the transmittance ($T$) data were converted into absorbance ($A$) (2.1) to obtain the same data representation of a reference commercial spectrometer device.

$$A = 2 - \log_{10}\left(100 \times \frac{T}{2^{16}}\right). \qquad (2.1)$$

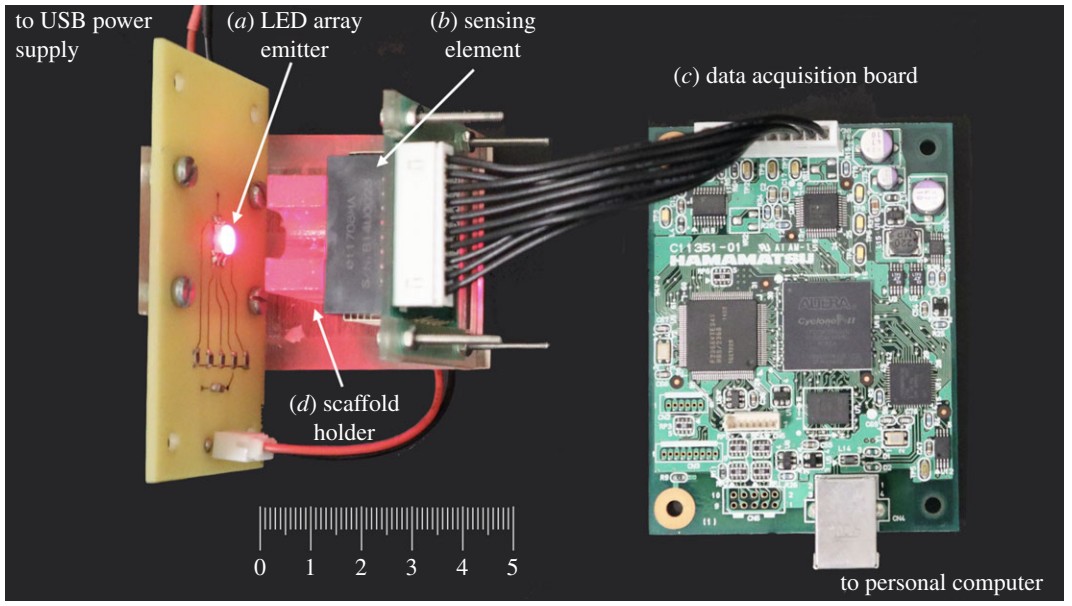

**Figure 2.** Photograph of the custom-made spectrometer device. (*a*) LED array emitter; (*b*) sensing element; (*c*) data acquisition board; (*d*) scaffold holder.

## 2.3. Commercial spectrometer device measurements

A commercial spectrometer device (Infinite M200, Tecan, Männedorf, Switzerland) was used as reference standard for the optical measurements. Scaffold absorbance was measured including each of them in a well of a standard 96-well plate. To not compromise the integrity of the samples, every single scaffold was placed over each well (figure 3*a*) and removing the underlying air with an insulin syringe (figure 3*b*) the perfect fitting was obtained (figure 3*c*).

The absorbance of each sample was then acquired with a 10 nm step within the same range (640–1000 nm) of the custom-made spectrometer device.

## 2.4. Statistical analysis

The non-parametric Mann–Whitney–Wilcoxon test was implemented using the Matlab (Mathworks, Natick, MA, USA) environment to assess significant differences among groups. The significance level was set at $p < 0.01$. All results are presented as mean ± standard deviation.

## 3. Results

Three-dimensional alginate hydrogel scaffolds at different $CaCO_3$ concentrations (20, 40, 80 mM) and time points (day 0, 4, 7, 11) were measured using the proposed custom-made device, and a commercial spectrometer (Infinite M200, Tecan) intended as reference standard to validate the prototype. To reduce noise effects, the initial analysis was performed using a 40 nm slider (i.e. moving average window with a centre wavelength ±20 nm), between 640 and 1000 nm.

The average absorbance spectra measured with the two devices are plotted in figure 4. The continuous and dotted lines represent the measurements obtained from the commercial and the custom-made devices, respectively. Their distinct shape is due to the different emitter light source of the two devices. In particular, the commercial device generates separately each specific wavelength of the spectrum, while the custom-made device is able to emit simultaneously only five specific wavelengths (troughs of the dash-dot line). Moreover, the two spectrometer devices have a different acquisition step (10 versus 20 nm). To make them uniform an interpolation was performed on the data acquired with the custom-made device to reconstruct the neighbourhood missing points. In this way, each point of the spectra identifies a specific wavelength in the 640–1000 nm range with a 10 nm step. The colour code helps to find the correspondence between the same wavelengths measured with the two devices.

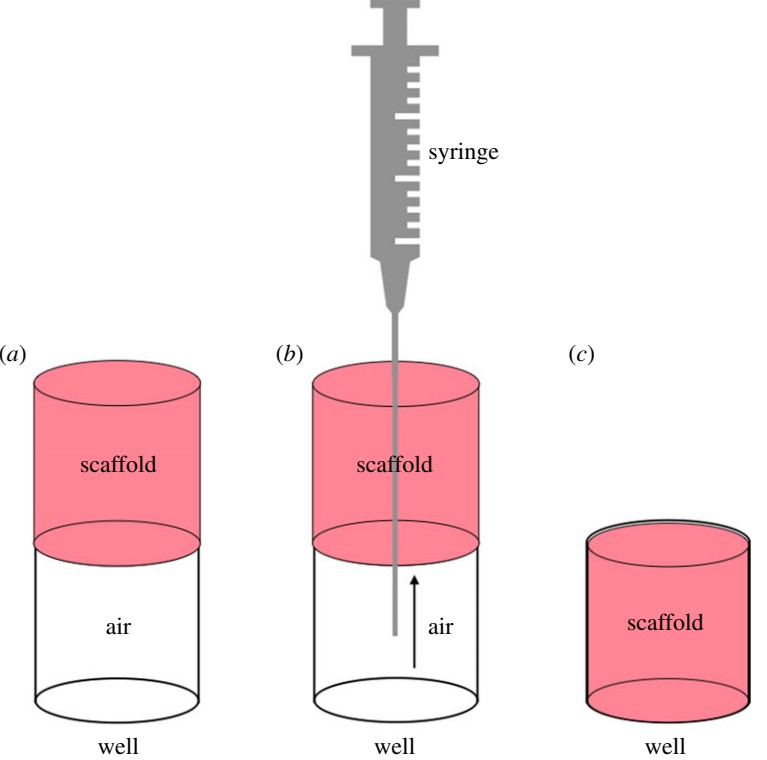

**Figure 3.** Scaffold inclusion method. Scaffold positioning over each well (*a*); removal of underlying air (*b*); scaffold/well fitting (*c*).

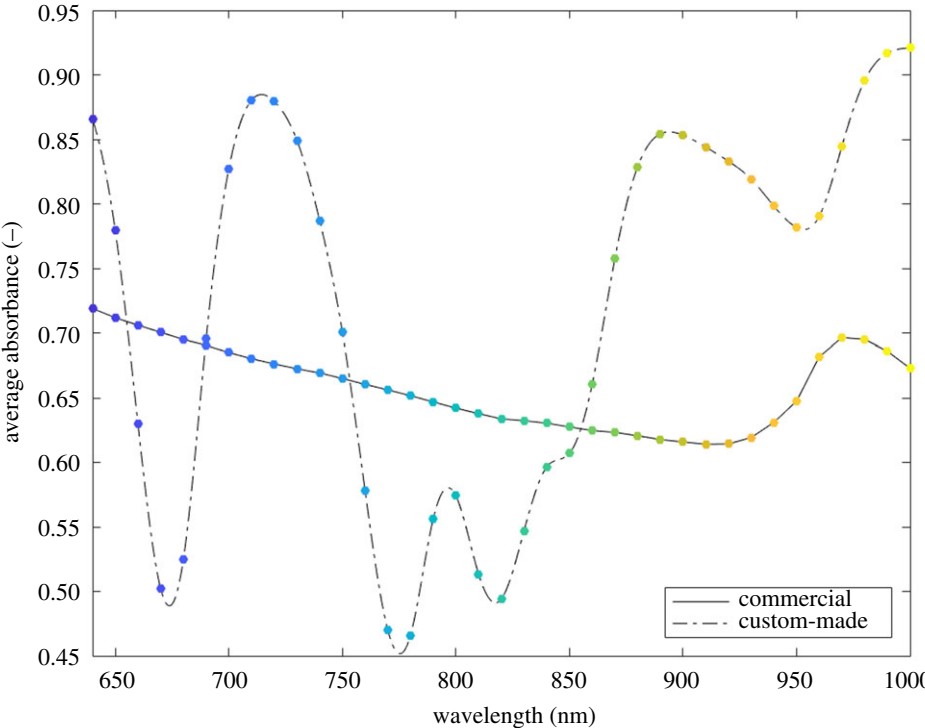

**Figure 4.** Average absorbance measured with the commercial (continuous line) and the custom-made (dashed-dotted line) devices. The points represent the discrete measured average absorbance values in the 640–1000 nm range with a 10 nm step. The colour code helps to find the correspondence between points measured at the same wavelengths with the two devices.

To analyse the trend of the measured spectra on samples with different amounts of calcium salt, a comparison of the median values ($n = 36$ for each wavelength) was performed (figure 5). The continuous (commercial device) and the dash-dot (custom-made device) lines are referring to scaffolds

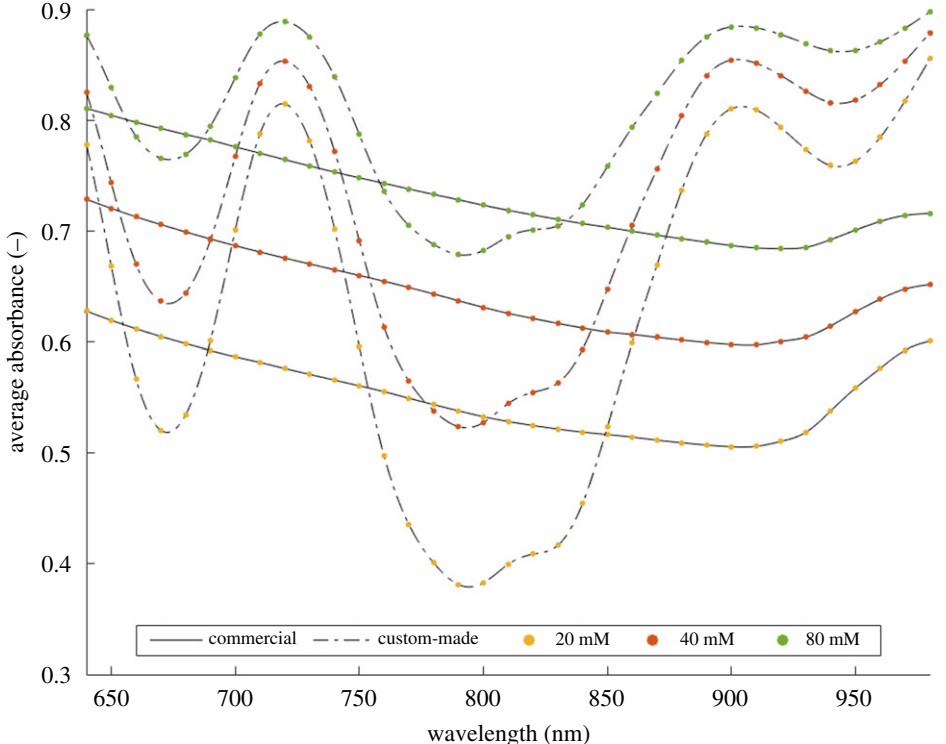

**Figure 5.** Comparison of sample median values among the different wavelengths. The continuous and dash-dot lines show the measured average absorbances, for the scaffold at 20 (yellow), 40 (red), 80 (green) mM CaCO$_3$, with the commercial and custom-made spectrometer devices, respectively.

at 20 (yellow), 40 (red) and 80 (green) mM CaCO$_3$. The trends highlight how both devices are able to distinguish the different CaCO$_3$ levels, maintaining at each wavelength a proportional distance for the increasing concentrations.

Considering this proportionality, a single wavelength was identified to simplify the following analysis. Its selection was performed calculating (figure 6) a similarity index ($S$) (equation 3.1) for each couple $x_1$, $x_2$ of points ($i$) at the same wavelength.

$$S = \frac{1}{1 + \sqrt{\sum_i (x_1 - x_2)^2}}. \tag{3.1}$$

In particular, this index allows comparison of the $x_1$ (commercial device) $x_2$ (custom-made device) points, despite the distinct shape of the spectra. The peak with the maximum value (0.996) of similarity was then chosen to define the reference wavelength at 690($\pm$20) nm.

Figure 7 (see also electronic supplementary material, figure S1) shows the average absorbance distribution measured with the commercial and custom-made spectrometer devices, at the selected wavelength. The box plot representation highlights how both devices are able to distinguish the scaffolds at the different CaCO$_3$ levels. Indeed, looking at the notches of each box, these are clearly not overlapped, thus the median of the different distribution can be considered as statistically different [19]. Moreover, the average absorbance distributions measured with the custom-made spectrometer device are more compact and uniform, thus a lower data dispersion is obtained [19]. This described behaviour was also observed in the other wavelength ranges (data not shown).

The above reported agreement between the measurements performed with the two spectrometer devices was further investigated as shown in figure 8. Linear regression and Bland–Altman tests show all the measures ($n = 108$) performed with the two spectrometer devices. In particular, the linear regression (figure 8a) analysis demonstrates an $r^2$ correlation coefficient equal to 0.78 as an index of agreement between our prototype and the reference device. This information was refined through a Bland–Altman test to provide an evaluation based on the difference between the two measurements [20]. Figure 8b highlights how the custom-made spectrometer systematically approximates the commercial device, with an overestimation limited to 0.0044. Moreover, the differences among the

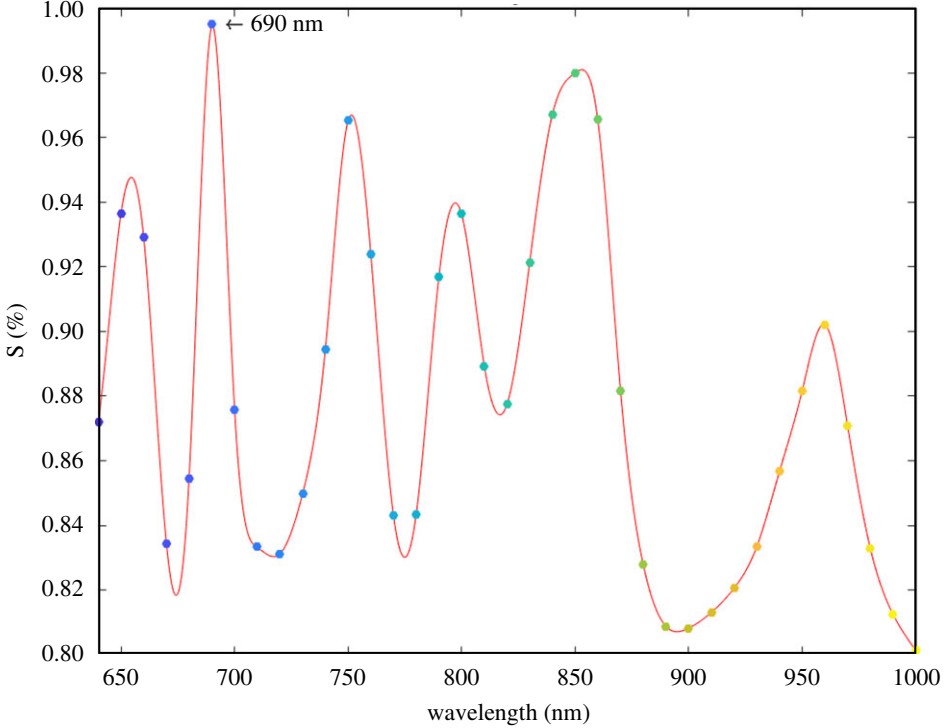

**Figure 6.** Similarity index between the couples of wavelengths measured with the two devices.

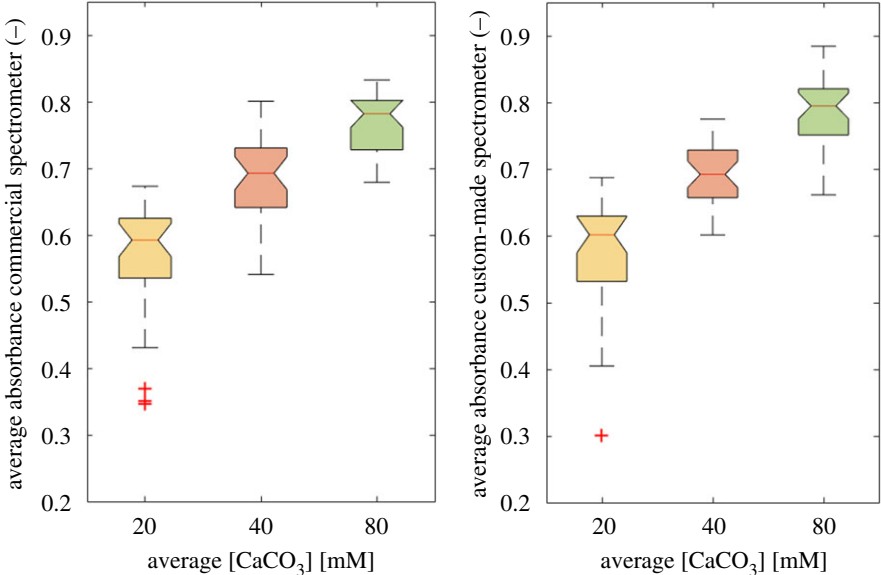

**Figure 7.** Average absorbance distribution at the selected wavelength. The panels show the average absorbance measured with the commercial (left) and the custom-made (right) spectrometer devices. A different colour code is used for each different $CaCO_3$ mM concentration (i.e. 20 = yellow, 40 = red and 80 = green). (+, outlier values).

measured data are almost all (90%) included within the confidence interval (lower limit −0.0208 and upper limit 0.0297). These results allow us to validate the custom-made spectrometer as able to reproduce the behaviour of the reference standard device.

Stability of the measurement method is evidenced when the validated custom-made spectrometer is used to evaluate the differential amounts of calcium salt in the alginate hydrogel scaffolds at different days.

Figure 9 shows the bar graph of scaffold measured values at different $CaCO_3$ levels (20, 40, 80 mM) and time points (day 0, 4, 7, 11), acquired with the commercial (left panel) and the custom-made (right

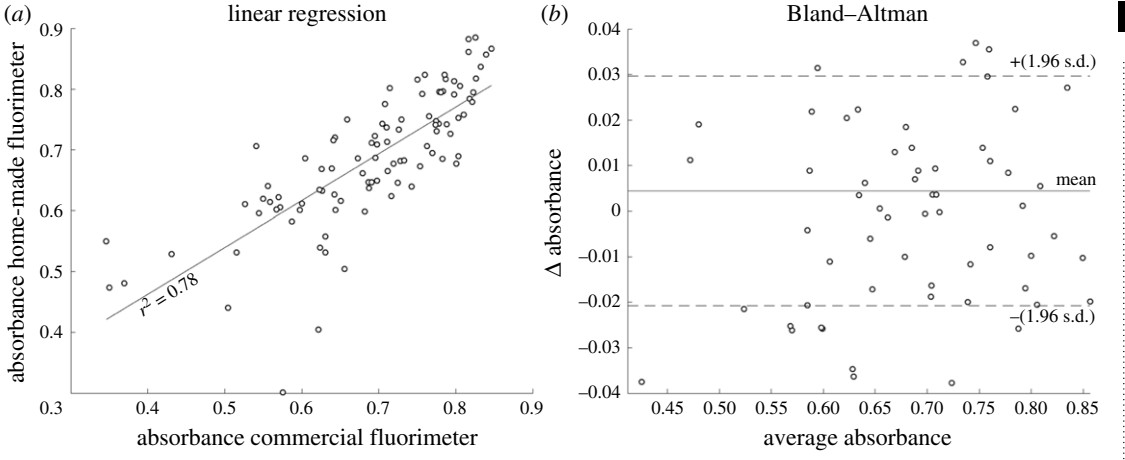

**Figure 8.** Level of agreement between the two spectrometer devices. (*a*) Linear regression; $r^2 = 0.78$. (*b*) Bland–Altman test; mean = 0.0044, lower limit = −0.0208, upper limit = 0.0297.

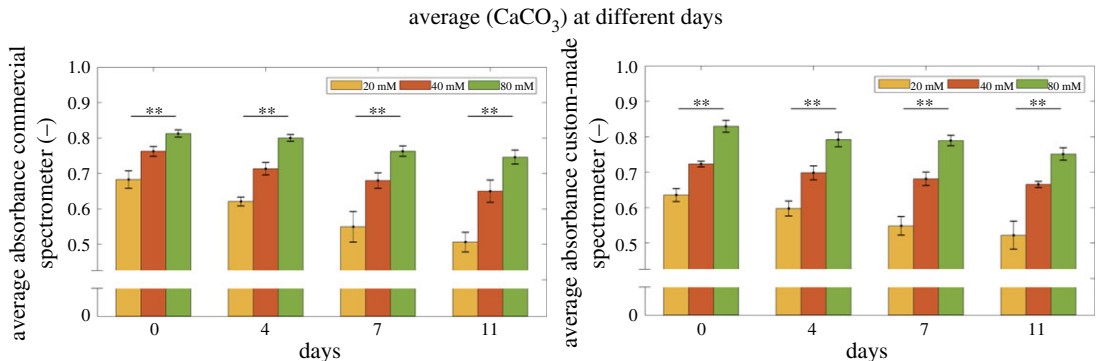

**Figure 9.** Average absorbance values measured in scaffolds with different $CaCO_3$ levels (20, 40, 80 mM) acquired at distinct time points (day 0, 4, 7, 11), with either the commercial (left panel) or the custom-made (right panel) spectrometers at $690 \pm 20$ nm. **Statistically significant difference (Mann–Whitney–Wilcoxon test; $p < 0.01$).

panel) spectrometers at $690 \pm 20$ nm. A Mann–Whitney–Wilcoxon test was applied when comparing intra- ($CaCO_3$ levels) and inter- (days) groups. All the intra-group values show a significant statistical difference ($p < 0.01$) among the different $CaCO_3$ levels, confirming the ability of both spectrometer devices to distinguish the different scaffold $CaCO_3$ content. On the other hand, all the inter-groups show a non-significant statistical difference among the measures performed at different days, and suggest the feasibility of stable assessment of mineral content at least in the measured time frame.

## 4. Discussion

During the last decade, scaffold-based TEed products have been developed in laboratories to better replicate *in vitro* the *in vivo* cell biology [21,22]. To this end, various biomaterials have been used as ECM substitutes to provide support for cell attachment, proliferation and differentiation.

Among these, hydrogel scaffolds have been widely used for their biological and mechanical properties. Their characteristics include non-immunogenicity, biocompatibility, non-toxicity and bioactivity [23]. Moreover, focalizing on specific TE approaches, such as the *in vitro* bone phenotype commitment, these templates must comply with other criteria, i.e. being osteoconductive and acting as a support for new bone deposition. In this respect, the ionic species associated with the biomaterial scaffold, such as calcium salts [24], play an important role during the bone ECM deposition and mineralization [25].

These biological processes are traditionally monitored via destructive techniques. As an example, to observe the bone ECM mineralization, histological assays are traditionally used. However, this method requires the fixation, sectioning and staining of the sample, preventing its monitoring over time. Consequently, for each experiment a large amount of scaffolds is needed, increasing costs and time.

To solve these issues, the use of non-destructive techniques can be considered as an alternative approach, leading also to more accurate and reproducible results [8]. Among them, spectroscopy-based methods stand out, since the transparency of the hydrogel scaffolds enables their use.

Starting from this paradigm, the aim of this study was to explore the reliability of measuring differential amounts of calcium salt in alginate hydrogel scaffolds for TE applications, by using a miniaturized and inexpensive custom-made spectrometer device. For this study three-dimensional alginate hydrogel scaffolds at different $CaCO_3$ concentrations (20, 40, 80 mM) and time points (day 0, 4, 7, 11) were measured by a custom-made spectrometer device. The same scaffolds were also measured by a commercial spectrometer device (Infinite M200, Tecan) used as reference standard.

The commercial device generates separately each specific wavelength of the spectrum, while the custom-made device is able to emit simultaneously at five specific wavelengths. For this reason a first exploratory measure was performed in order to analyse the output of the spectrometer devices. Due the different emitter light source, two average absorbance spectra with a distinct shape were obtained (figure 4). Therefore, to allow the comparison of the devices, a specific analysis was performed. The trends in figure 5 show at each wavelength a proportional distance of the average absorbance spectra for the increasing $CaCO_3$ concentrations. Considering this proportionality, a similarity index (figure 6) was computed to allow the comparison. The wavelength (690 nm) with the highest similarity value was then used for further analysis.

The box plot representation in figure 7 (see also electronic supplementary material, figure S1) highlights the ability of both the spectrometer devices to distinguish the scaffolds at the different $CaCO_3$ levels. Moreover, less data dispersion is observed when the custom-made spectrometer device is used. To confirm the agreement between the measurements performed with the two spectrometer devices a linear regression and a Bland–Altman test (figure 8) were performed: (i) a correlation coefficient $r^2 = 0.78$, (ii) the ability of the custom-made spectrometer to systematically approximate the commercial device, (iii) the differences among the measured data (almost 90%) included within the confidence interval, allow us to validate the custom-made spectrometer as able to reproduce the behaviour of the reference standard device.

In addition, the stability of the measurement method is evidenced when the validated custom-made spectrometer is used to evaluate the differential amounts of calcium salt in the alginate hydrogel scaffolds at different days. Indeed, comparing the intra- ($CaCO_3$ levels) and the inter- (days) groups values (figure 9), a significant statistical difference ($p < 0.01$) among the different $CaCO_3$ levels and a non-significant statistical difference among the measures performed at different days were observed. These data confirm the ability of the custom-made spectrometer to distinguish the different scaffold $CaCO_3$ content, at same extent of the reference device.

## 5. Conclusion

This study shows the feasibility of a light spectroscopy-based method as useful for the non-destructive analysis of TEed constructs. The obtained results demonstrate the ability of our custom-made spectrometer device to measure the differential amounts of calcium salt in alginate hydrogel scaffolds. Such evidence asks for a future application of this device to monitor *in vitro* early stages of bone ECM matrix mineralization. In this respect, future experiments are planned using stem cells induced to differentiate towards the osteogenic phenotype on board of both alginate and other biopolymer-based hydrogel scaffolds. Each measurement was performed without requiring elaborate sample preparation and/or a complex instrumentation, such as for FTIR and Raman spectroscopy.

Based on these results the custom-made spectrometer device appears as a useful option to perform real-time/in-line analysis of TEed constructs. Its cost and performance make it competitive with other recently proposed non-destructive approaches [26]. Furthermore, the small dimensions allow us to consider the device as a component that can be easily integrated on board of recently prototyped bioreactor systems [27–31], for the monitoring of the TEed constructs during their conditioning.

Data accessibility. Data are available within Dryad Digital Repository: https://doi.org/10.5061/dryad.gxd2547m3 [32].
Authors' contributions. J.L., V.B. and M.C. performed the experiments. J.L. and V.B. evaluated their results and prepared the manuscript, including text and figures. E.R. developed the custom-made spectrometer device. E.G. and S.S. conceived the study and contributed to data analysis. All the authors critically read, edited and approved the manuscript.
Competing interests. We declare we have no competing interests.

Funding. This work was also supported in part by the Italian Ministry for Education, University and Research (MIUR) under the program 'Dipartimenti di Eccellenza (2018–2022)'.

Acknowledgements. The authors gratefully acknowledge financial support from Fondazione Cassa di Risparmio in Bologna.

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
