## [Peer Review File · Royal Society Open Science]

Review History

RSOS-210791.R0 (Original submission)

Review form: Reviewer 1

Is the manuscript scientifically sound in its present form?

No

Are the interpretations and conclusions justified by the results?

Yes

Is the language acceptable?

Yes

Do you have any ethical concerns with this paper?

No

Have you any concerns about statistical analyses in this paper?

No

Recommendation?

Major revision is needed (please make suggestions in comments)

Comments to the Author(s)

This study is potentially significant as it explores the reliability of “a miniaturized and inexpensive custom-made spectrometer device” for measuring differential amounts of a calcium salt in an alginate hydrogel. This system would enable a “real-time, in-line, in situ analysis,” that could be used in tissue engineering studies (e.g., non-destructive characterization of the mineral content in polymeric scaffolds, bioreactor systems, etc.). The following comments have been provided for the authors’ consideration and could improve the manuscript.

1) Page 2: The introduction is very brief. Some background information on the current state of the art and a comparison with similar studies reported by others would be helpful to readers.

2) Page 3: The suppliers of all chemicals should be identified. This includes RPMI, used under “3D Alginate Hydrogel scaffolds”. Also, letters “A” and “T” in Eq. 1 should be defined.

3) Page 3: The information provided under “Custom-made spectrometer device development and measurements” may not be adequate for a reader to replicate the study and build the device. Additional details and a more descriptive image (Figure 1) would be necessary.

4) Page 10 and Figure 8: According to the manuscript, the stability of the “measurement method is evidenced when the validated custom-made spectrometer is used to evaluate the differential amounts of calcium salt in the alginate hydrogel scaffolds at different days.” Unlike the previous results (Figures 3 – 7), Figure 8 only shows the results of the custom-made device. It would be desirable to see the results at the different time points coming from the commercial device.

5) Page 11: The manuscript shows a short paragraph at the end, summarizing the observations of this study. Nevertheless, it would be important to have distinct sections for the discussion and/or conclusions, including an overview of the limitations, future directions, and additional potential applications beyond the results shown in this study.

6) The manuscript does not show any image of the hydrogels prepared in this study (neither optical nor scanning electron microscopy).

7) Overall, the manuscript is short. Additional characterizations using a different technique to demonstrate a uniform distribution of the calcium salt within these hydrogels seem to be essential for this study.

Review form: Reviewer 2

Is the manuscript scientifically sound in its present form?

Yes

Are the interpretations and conclusions justified by the results?

Yes

Is the language acceptable?

Yes

Do you have any ethical concerns with this paper?

No

Have you any concerns about statistical analyses in this paper?

No

Recommendation?

Accept with minor revision (please list in comments)

Comments to the Author(s)

The article is well designed and written. Please make the following corrections before accepting

1. Summarize the article abstract in one paragraph
2. Find out more about the calcium phosphate family. You can get help from the following sources: <https://doi.org/10.1016/j.bsecv.2020.05.001>, <https://doi.org/10.1016/j.bsecv.2017.05.001>
3. More details are needed about hydrogels and gels. You can get help from the following sources: <https://doi.org/10.1016/j.matchemphys.2019.122305>, <https://doi.org/10.1021/acsbiomaterials.9b01689>
4. Explain Equation 1 further and clarify what each parameter specifies.
5. If possible, take a few microscopic images of the scaffolds and add them to the manuscript
6. The number of references is too low, please add more studies

Decision letter (RSOS-210791.R0)

Dear Dr Lovecchio

The Editors assigned to your paper RSOS-210791 "Design of a custom-made device for real-time optical measurement of differential mineral concentrations in 3D scaffolds for bone tissue engineering" have now received comments from reviewers and would like you to revise the paper in accordance with the reviewer comments and any comments from the Editors. Please note this decision does not guarantee eventual acceptance.

Please submit your revised manuscript and required files (see below) no later than 21 days from today's (ie 08-Sep-2021) date. Note: the ScholarOne system will 'lock' if submission of the revision is attempted 21 or more days after the deadline. If you do not think you will be able to meet this deadline please contact the editorial office immediately.

on behalf of Dr Peter Munro (Associate Editor) and R. Kerry Rowe (Subject Editor)
openscience@royalsociety.org

Associate Editor Comments to Author (Dr Peter Munro):

Associate Editor: 1

Comments to the Author:

The reviewers have raised a number of queries which should be addressed before this manuscript can be published.

Reviewer comments to Author:

Reviewer: 1

Comments to the Author(s)

This study is potentially significant as it explores the reliability of “a miniaturized and inexpensive custom-made spectrometer device” for measuring differential amounts of a calcium salt in an alginate hydrogel. This system would enable a “real-time, in-line, in situ analysis,” that could be used in tissue engineering studies (e.g., non-destructive characterization of the mineral content in polymeric scaffolds, bioreactor systems, etc.). The following comments have been provided for the authors’ consideration and could improve the manuscript.

1) Page 2: The introduction is very brief. Some background information on the current state of the art and a comparison with similar studies reported by others would be helpful to readers.

2) Page 3: The suppliers of all chemicals should be identified. This includes RPMI, used under “3D Alginate Hydrogel scaffolds”. Also, letters “A” and “T” in Eq. 1 should be defined.

3) Page 3: The information provided under “Custom-made spectrometer device development and measurements” may not be adequate for a reader to replicate the study and build the device. Additional details and a more descriptive image (Figure 1) would be necessary.

4) Page 10 and Figure 8: According to the manuscript, the stability of the “measurement method is evidenced when the validated custom-made spectrometer is used to evaluate the differential amounts of calcium salt in the alginate hydrogel scaffolds at different days.” Unlike the previous results (Figures 3 – 7), Figure 8 only shows the results of the custom-made device. It would be desirable to see the results at the different time points coming from the commercial device.

- 5) Page 11: The manuscript shows a short paragraph at the end, summarizing the observations of this study. Nevertheless, it would be important to have distinct sections for the discussion and/or conclusions, including an overview of the limitations, future directions, and additional potential applications beyond the results shown in this study.
- 6) The manuscript does not show any image of the hydrogels prepared in this study (neither optical nor scanning electron microscopy).
- 7) Overall, the manuscript is short. Additional characterizations using a different technique to demonstrate a uniform distribution of the calcium salt within these hydrogels seem to be essential for this study.

Reviewer: 2

Comments to the Author(s)

The article is well designed and written. Please make the following corrections before accepting

1. Summarize the article abstract in one paragraph
2. Find out more about the calcium phosphate family. You can get help from the following sources: <https://doi.org/10.1016/j.bsecv.2020.05.001>, <https://doi.org/10.1016/j.bsecv.2017.05.001>
3. More details are needed about hydrogels and gels. You can get help from the following sources: <https://doi.org/10.1016/j.matchemphys.2019.122305>, <https://doi.org/10.1021/acsbio.2019.01689>
4. Explain Equation 1 further and clarify what each parameter specifies.
5. If possible, take a few microscopic images of the scaffolds and add them to the manuscript
6. The number of references is too low, please add more studies

===PREPARING YOUR MANUSCRIPT===

If you have been asked to revise the written English in your submission as a condition of publication, you must do so, and you are expected to provide evidence that you have received language editing support. The journal would prefer that you use a professional language editing

service and provide a certificate of editing, but a signed letter from a colleague who is a native speaker of English is acceptable. Note the journal has arranged a number of discounts for authors using professional language editing services (<https://royalsociety.org/journals/authors/benefits/language-editing/>).

===PREPARING YOUR REVISION IN SCHOLARONE===

-- If you have uploaded ESM files, please ensure you follow the guidance at <https://royalsociety.org/journals/authors/author-guidelines/#supplementary-material> to include a suitable title and informative caption. An example of appropriate titling and captioning

may be found at https://figshare.com/articles/Table_S2_from_Is_there_a_trade-off_between_peak_performance_and_performance_breadth_across_temperatures_for_aerobic_sc_ope_in_teleost_fishes_/3843624.

Author's Response to Decision Letter for (RSOS-210791.R0)

See Appendix A.

RSOS-210791.R1 (Revision)

Review form: Reviewer 1

Is the manuscript scientifically sound in its present form?

Yes

Are the interpretations and conclusions justified by the results?

Yes

Is the language acceptable?

Yes

Do you have any ethical concerns with this paper?

No

Have you any concerns about statistical analyses in this paper?

No

Recommendation?

Accept with minor revision (please list in comments)

Comments to the Author(s)

The manuscript has been adequately revised to address the concerns raised by the reviewers. Nevertheless, the authors are encouraged to proof-read their revised manuscript before resubmission. Some corrections are necessary before consideration by the journal. Here are some examples:

1) Page 2: In the following sentence, “[Ref]” appears to be a place-holder for a missing reference number, or it is possibly a typographical error.

This test conveniently reports about bone mineral content (BMC) and density (BMD) in patients, although requiring dedicated expensive and bulky instruments [Ref] inappropriate for a cellular study in a laboratory.

2) Page 2: Likewise, it is unclear if “[Vanderoost 14]” in the following sentence is a reference citation, which seems to be in an unusual format:

Among those, X-ray uCT was demonstrated as a powerful non-destructive imaging technique to obtain, with the highest resolution, precise quantitative and qualitative information on the bone tissue microstructure again, without the need for conventional destructive methods [Vanderoost 14].

3) Table 1: The table caption should be placed above the table.

4) Figure 9: The caption does not clearly describe the two graphs. It is unclear which graph the caption is referring to. These graphs should contain labels (a, b) and a distinctive caption for each. According to the authors’ response to reviewers, one of the graphs is presenting the newly-added results, but the caption has not been updated accordingly. The response to reviewers states “We added in old Figure 8 (now Figure 9, left panel) the results collected with the commercial device at the different time points. Results from the custom-made device appear in the right panel of this same Figure.”

5) Page 11: The following sentence should be a part of the previous paragraph or the next paragraph. It is currently showing as a single-sentence paragraph. “A single wavelength (690nm) was identified aiming at simplify any subsequent analysis.” In addition, this sentence needs to be rephrased (“aiming at simplify” is unclear).

6) Page 12: The same issue can be seen under Conclusion. There are three single-sentenced paragraphs under this section.

Review form: Reviewer 2

Is the manuscript scientifically sound in its present form?

Yes

Are the interpretations and conclusions justified by the results?

Yes

Is the language acceptable?

Yes

Do you have any ethical concerns with this paper?

No

Have you any concerns about statistical analyses in this paper?

No

Recommendation?

Accept as is

Comments to the Author(s)

Accept

Decision letter (RSOS-210791.R1)

Dear Dr Lovecchio

On behalf of the Editors, we are pleased to inform you that your Manuscript RSOS-210791.R1 "Design of a custom-made device for real-time optical measurement of differential mineral concentrations in 3D scaffolds for bone tissue engineering" has been accepted for publication in Royal Society Open Science subject to minor revision in accordance with the referees' reports. Please find the referees' comments along with any feedback from the Editors below my signature.

Please submit your revised manuscript and required files (see below) no later than 7 days from today's (ie 29-Oct-2021) date. Note: the ScholarOne system will 'lock' if submission of the revision is attempted 7 or more days after the deadline. If you do not think you will be able to meet this deadline please contact the editorial office immediately.

on behalf of Dr Peter Munro (Associate Editor) and R. Kerry Rowe (Subject Editor)
openscience@royalsociety.org

Associate Editor Comments to Author (Dr Peter Munro):

Thank you for revising your manuscript. I am pleased to recommend that this paper be published once minor revisions raised by Reviewer 2 have been addressed.

Reviewer comments to Author:

Reviewer: 1

Comments to the Author(s)

The manuscript has been adequately revised to address the concerns raised by the reviewers. Nevertheless, the authors are encouraged to proof-read their revised manuscript before

resubmission. Some corrections are necessary before consideration by the journal. Here are some examples:

1) Page 2: In the following sentence, “[Ref]” appears to be a place-holder for a missing reference number, or it is possibly a typographical error.

This test conveniently reports about bone mineral content (BMC) and density (BMD) in patients, although requiring dedicated expensive and bulky instruments [Ref] inappropriate for a cellular study in a laboratory.

2) Page 2: Likewise, it is unclear if “[Vandroost 14]” in the following sentence is a reference citation, which seems to be in an unusual format:

Among those, X-ray uCT was demonstrated as a powerful non-destructive imaging technique to obtain, with the highest resolution, precise quantitative and qualitative information on the bone tissue microstructure again, without the need for conventional destructive methods [Vandroost 14].

3) Table 1: The table caption should be placed above the table.

4) Figure 9: The caption does not clearly describe the two graphs. It is unclear which graph the caption is referring to. These graphs should contain labels (a, b) and a distinctive caption for each. According to the authors’ response to reviewers, one of the graphs is presenting the newly-added results, but the caption has not been updated accordingly. The response to reviewers states “We added in old Figure 8 (now Figure 9, left panel) the results collected with the commercial device at the different time points. Results from the custom-made device appear in the right panel of this same Figure.”

5) Page 11: The following sentence should be a part of the previous paragraph or the next paragraph. It is currently showing as a single-sentence paragraph. “A single wavelength (690nm) was identified aiming at simplify any subsequent analysis.” In addition, this sentence needs to be rephrased (“aiming at simplify” is unclear).

6) Page 12: The same issue can be seen under Conclusion. There are three single-sentenced paragraphs under this section.

Reviewer: 2

Comments to the Author(s)

Accept

===PREPARING YOUR MANUSCRIPT===

one version should clearly identify all the changes that have been made (for instance, in coloured highlight, in bold text, or tracked changes);

===PREPARING YOUR REVISION IN SCHOLARONE===

- If you are providing image files for potential cover images, please upload these at this step, and inform the editorial office you have done so. You must hold the copyright to any image provided.
- A copy of your point-by-point response to referees and Editors. This will expedite the preparation of your proof.

- Ensure that your data access statement meets the requirements at <https://royalsociety.org/journals/authors/author-guidelines/#data>. You should ensure that you cite the dataset in your reference list. If you have deposited data etc in the Dryad repository, please only include the 'For publication' link at this stage. You should remove the 'For review' link.
- If you are requesting an article processing charge waiver, you must select the relevant waiver option (if requesting a discretionary waiver, the form should have been uploaded, see 'File upload' above).
- If you have uploaded any electronic supplementary (ESM) files, please ensure you follow the guidance at <https://royalsociety.org/journals/authors/author-guidelines/#supplementary-material> to include a suitable title and informative caption. An example of appropriate titling and captioning may be found at https://figshare.com/articles/Table_S2_from_Is_there_a_trade-off_between_peak_performance_and_performance_breadth_across_temperatures_for_aerobic_scope_in_teleost_fishes_/3843624.

Author's Response to Decision Letter for (RSOS-210791.R1)

See Appendix B.

Decision letter (RSOS-210791.R2)

Dear Dr Lovecchio,

It is a pleasure to accept your manuscript entitled "Design of a custom-made device for real-time optical measurement of differential mineral concentrations in 3D scaffolds for bone tissue engineering" in its current form for publication in Royal Society Open Science. The comments of the reviewer(s) who reviewed your manuscript are included at the foot of this letter.

on behalf of Dr Peter Munro (Associate Editor) and R. Kerry Rowe (Subject Editor)
openscience@royalsociety.org

Appendix A

Joseph Lovecchio
Laboratory of Cellular and Molecular Engineering "Silvio Cavalcanti"
Department of Electrical, Electronic and Information Engineering "Guglielmo Marconi" (DEI)
University of Bologna, Via dell'università 50, 47522 Cesena (FC), Italy
e-mail: joseph.lovecchio@unibo.it tel: +390547338953

To the kind attention of
Royal Society Open Science

Dear Editor,

we thank you for considering the manuscript entitled "Design of a custom-made device for real-time optical measurement of differential mineral concentrations in 3D scaffolds for bone tissue engineering" by Joseph Lovecchio, Valentina Betti, Marilisa Cortesi, Enrico Ravagli, Stefano Severi, Emanuele Giordano, intended as a regular submission as a Research Article for Royal Society Open Science.

We appreciate the time you have dedicated to our paper and the valuable comments from the Reviewers.

We did our best to answer in detail to all the questions raised by the Reviewers and to reformat the manuscript accordingly.

Please find uploaded to Scholarone application:

- a .zip file (Lovecchio_Betti_2021_RSOS_(Rev)) containing a copy of the revised manuscript (new text: red, deleted text: black strikethrough);
- a .zip file (Lovecchio_Betti_2021_RSOS_(Final)) containing a copy of the 'clean' version of the new manuscript.

Point-by-point answers to the Reviewers' comments are presented in the following pages.

We hope that our careful consideration of the Reviewers' concerns - as far as possible within the time allotted to report our answers (deadline October 1st, 2021) - increased the value of our manuscript to the readership of Royal Society Open Science and thank you for the opportunity to consider its resubmitted version.

We are looking forward to hearing from you.

Yours sincerely,
Dr. Joseph Lovecchio

#Reviewer 1

Comments to the Author(s)

This study is potentially significant as it explores the reliability of “a miniaturized and inexpensive custom-made spectrometer device” for measuring differential amounts of a calcium salt in an alginate hydrogel. This system would enable a “real-time, in-line, in situ analysis,” that could be used in tissue engineering studies (e.g., non-destructive characterization of the mineral content in polymeric scaffolds, bioreactor systems, etc.). The following comments have been provided for the authors’ consideration and could improve the manuscript.

We thank the Reviewer for the kind appreciation of our work and the specific comments to improve the manuscript.

1) Page 2: The introduction is very brief. Some background information on the current state of the art and a comparison with similar studies reported by others would be helpful to readers.

As requested, we added to the Introduction section some background information on the current state-of-the-art about measurement of bone mineral content/density. Dual-energy x-ray absorptiometry (DXA)-based assays in patients for clinical purposes are mentioned. X-ray μ CT-based non-destructive techniques have also been reported. Both these approaches however look inappropriate (expensive and bulky) for cell laboratory TE applications.

The basis for our light spectroscopy-based proposed approach were mentioned in the manuscript. To the best of our knowledge, we did not retrieve in the literature other compact and cheap devices to compare with ours.

2) Page 3: The suppliers of all chemicals should be identified. This includes RPMI, used under “3D Alginate Hydrogel scaffolds”. Also, letters “A” and “T” in Eq. 1 should be defined.

We have amended the manuscript as requested.

3) Page 3: The information provided under “Custom-made spectrometer device development and measurements” may not be adequate for a reader to replicate the study and build the device. Additional details and a more descriptive image (Figure 1) would be necessary.

The information provided under “Custom-made spectrometer device development and measurements” has been reformulated in conjunction with providing a picture of the real device (new Figure 2) in order to support a reader willing to build the device and replicate the study.

4) Page 10 and Figure 8: According to the manuscript, the stability of the “measurement method is evidenced when the validated custom-made spectrometer is used to evaluate the differential amounts of calcium salt in the alginate hydrogel scaffolds at different days.” Unlike the previous results (Figures 3 – 7), Figure 8 only shows the results of the custom-made device. It would be desirable to see the results at the different time points coming from the commercial device.

We added in old Figure 8 (now Figure 9, left panel) the results collected with the commercial device at the different time points. Results from the custom-made device appear in the right panel of this same Figure.

5) Page 11: The manuscript shows a short paragraph at the end, summarizing the observations of this study. Nevertheless, it would be important to have distinct sections for the discussion and/or conclusions, including an overview of the limitations, future directions, and additional potential applications beyond the results shown in this study.

We added two new sections (i.e.: discussion and conclusion) to the manuscript. The conclusion section includes an overview of the limitations, future directions, and additional potential applications beyond the results shown in this study.

6) The manuscript does not show any image of the hydrogels prepared in this study (neither optical nor scanning electron microscopy).

A new Figure 1 shows an image of 3D hydrogel scaffolds at the three different CaCO₃ concentrations (20, 40, and 80 mM) evaluated in this study. Accordingly, the 3D alginate hydrogel scaffold paragraph was reformulated in the “Material and Methods” section, including a new Table (Table 1) detailing the reagent mix used to cast them.

7) Overall, the manuscript is short. Additional characterizations using a different technique to demonstrate a uniform distribution of the calcium salt within these hydrogels seem to be essential for this study.

In the section “Supplementary material” a new figure (S2) shows an image of Alizarin red-stained slices from hydrogel scaffold at different CaCO₃ concentrations (20, 40, and 80 mM) evaluated in this study.

#Reviewer 2

Comments to the Author(s)

The article is well designed and written. Please make the following corrections before accepting

We thank the Reviewer for the kind appreciation of our work and the specific comments to improve the manuscript.

1) Summarize the article abstract in one paragraph

The abstract was amended as requested.

2) Find out more about the calcium phosphate family. You can get help from the following sources: <https://doi.org/10.1016/j.bsecv.2020.05.001>, <https://doi.org/10.1016/j.bsecv.2017.05.001>

The following reference has been added in a newly added Discussion section:

<https://doi.org/10.1016/j.bsecv.2017.05.001>

3) More details are needed about hydrogels and gels. You can get help from the following sources: <https://doi.org/10.1016/j.matchemphys.2019.122305>, <https://doi.org/10.1021/acsbiomaterials.9b01689>

The following references have been added in a newly added Discussion section:

<https://doi.org/10.1016/j.matchemphys.2019.122305>

<https://dx.doi.org/10.1021/acs.bioconjchem.0c00270>

<https://doi.org/10.1021/acsbiomaterials.9b01689>

4) Explain Equation 1 further and clarify what each parameter specifies.

We have amended the manuscript as requested.

5) If possible, take a few microscopic images of the scaffolds and add them to the manuscript

A new Figure 1 shows an image of 3D hydrogel scaffolds at the three different CaCO₃ concentrations (20, 40, and 80 mM) evaluated in this study. Accordingly, the 3D alginate hydrogel scaffold paragraph was reformulated in the “Material and Methods” section, including a new Table (Table 1) detailing the reagent mix used to cast them.

A new figure (S2) in the “Supplementary Material” sections shows an image of Alizarin red-stained slices from hydrogel scaffold at different CaCO₃ concentrations (20, 40, and 80 mM) evaluated in this study.

6) The number of references is too low, please add more studies

We have amended the manuscript as requested.

Appendix B

Joseph Lovecchio
Laboratory of Cellular and Molecular Engineering "Silvio Cavalcanti"
Department of Electrical, Electronic and Information Engineering "Guglielmo Marconi" (DEI)
University of Bologna, Via dell'università 50, 47522 Cesena (FC), Italy
e-mail: joseph.lovecchio@unibo.it tel: +390547338953

To the kind attention of
Royal Society Open Science

Dear Editor,

we thank you for considering the manuscript entitled "Design of a custom-made device for real-time optical measurement of differential mineral concentrations in 3D scaffolds for bone tissue engineering" by Joseph Lovecchio, Valentina Betti, Marilisa Cortesi, Enrico Ravagli, Stefano Severi, Emanuele Giordano, intended as a regular submission as a Research Article for Royal Society Open Science.

We appreciate the time you have dedicated to our paper and the valuable comments from the Reviewers.

We did our best to answer in detail to all the questions raised by the Reviewers and to reformat the manuscript accordingly.

Please find uploaded to Scholarone application:

- a .zip file (Lovecchio_Betti_2021_RSOS_(Rev2)) containing a copy of the revised manuscript (new text: red, deleted text: black strikethrough);
- a .zip file (Lovecchio_Betti_2021_RSOS_(Final_Rev2)) containing a copy of the 'clean' version of the new manuscript.

Point-by-point answers to the Reviewers' comments are presented in the following pages.

We hope that our careful consideration of the Reviewers' concerns - as far as possible within the time allotted to report our answers (deadline November 7th, 2021) - increased the value of our manuscript to the readership of Royal Society Open Science and thank you for the opportunity to consider its resubmitted version.

We are looking forward to hearing from you.

Yours sincerely,
Dr. Joseph Lovecchio

#Reviewer 1

Comments to the Author(s)

The manuscript has been adequately revised to address the concerns raised by the reviewers. Nevertheless, the authors are encouraged to proof-read their revised manuscript before resubmission. Some corrections are necessary before consideration by the journal. Here are some examples:

We thank the Reviewer for the kind appreciation of our work and the specific comments to improve the manuscript.

1) Page 2: In the following sentence, “[Ref]” appears to be a place-holder for a missing reference number, or it is possibly a typographical error.

This test conveniently reports about bone mineral content (BMC) and density (BMD) in patients, although requiring dedicated expensive and bulky instruments [Ref] inappropriate for a cellular study in a laboratory.

We have amended the manuscript as requested.

2) Page 2: Likewise, it is unclear if “[Vanderoost 14]” in the following sentence is a reference citation, which seems to be in an unusual format:

Among those, X-ray uCT was demonstrated as a powerful non-destructive imaging technique to obtain, with the highest resolution, precise quantitative and qualitative information on the bone tissue microstructure again, without the need for conventional destructive methods [Vanderoost 14].

We have amended the manuscript as requested.

3) Table 1: The table caption should be placed above the table.

We have amended the manuscript as requested.

4) Figure 9: The caption does not clearly describe the two graphs. It is unclear which graph the caption is referring to. These graphs should contain labels (a, b) and a distinctive caption for each. According to the authors’ response to reviewers, one of the graphs is presenting the newly-added results, but the caption has not been updated accordingly. The response to reviewers states “We added in old Figure 8 (now Figure 9, left panel) the results collected with the commercial device at the different time points. Results from the custom-made device appear in the right panel of this same Figure.”

Figure 9 caption has been reworded to properly convey its meaning to the readers.

5) Page 11: The following sentence should be a part of the previous paragraph or the next paragraph. It is currently showing as a single-sentence paragraph. “A single wavelength (690nm) was identified aiming at simplify any subsequent analysis.” In addition, this sentence needs to be rephrased (“aiming at simplify” is unclear).

The mentioned sentence was reworded as: “The wavelength (690nm) with the highest similarity value was then used for further analysis”.

6) Page 12: The same issue can be seen under Conclusion. There are three single-sentenced paragraphs under this section.

We have compacted the Conclusion section as requested.

#Reviewer 2

Comments to the Author(s)

Accept.

We thank the Reviewer for the kind appreciation of our work.